# Feedback of peripheral saccade targets to early foveal cortex

**Luca Kämmer[1,2,3]\*, Lisa M Kroell[2], Tomas Knapen[4,5,6], Martin Rolfs[2,7,8,9,10], Martin N Hebart[1,3,11]**

[1]Vision and Computational Cognition Group, Max Planck Institute of Human Cognitive and Brain Sciences, Leipzig, Germany; [2]Department of Psychology, Humboldt University Berlin, Berlin, Germany; [3]Department of Medicine, Justus Liebig University Giessen, Giessen, Germany; [4]Spinoza Center for Neuroimaging, KNAW Netherlands, Amsterdam, Netherlands; [5]Computational Cognitive Neuroscience and Neuroimaging, Netherlands Institute for Neuroscience, Amsterdam, Netherlands; [6]Experimental and Applied Psychology, Vrije University Amsterdam, Amsterdam, Netherlands; [7]Berlin School of Mind and Brain, Humboldt University Berlin, Berlin, Germany; [8]Exzellenzcluster Science of Intelligence, Technical University Berlin, Berlin, Germany; [9]Bernstein Center for Computational Neuroscience Berlin, Berlin, Germany; [10]Bernstein Center for Computational Neuroscience Berlin, Berlin, Germany; [11]Center for Mind, Brain and Behavior, Universities of Marburg, Giessen, and Darmstadt, Germany, Marburg, Germany

**\*For correspondence:**
kaemmer@cbs.mpg.de

## eLife Assessment

This **valuable** study addresses a question related to how we achieve visual stability across saccadic eye movements. The authors' gaze-contingent fMRI design provides **convincing** evidence that peripherally presented visual stimuli are represented in foveal visual cortex prior to a saccade. The results will be of interest to vision scientists and behavioural neuroscientists.

**Abstract** Human vision is characterized by frequent eye movements and constant shifts in visual input, yet our perception of the world remains remarkably stable. Here, we directly demonstrate image-specific foveal feedback to primary visual cortex in the context of saccadic eye movements. To this end, we used a gaze-contingent fMRI paradigm, in which peripheral saccade targets disappeared before they could be fixated. Despite no direct foveal stimulation, we were able to decode peripheral saccade targets from foveal retinotopic areas, demonstrating that image-specific feedback during saccade preparation may underlie this effect. Decoding was sensitive to shape but not semantic category of natural images, indicating feedback of only low-to-mid-level information. Cross-decoding to a control condition with foveal stimulus presentation indicates a shared representational format between foveal feedback and direct stimulation. Moreover, eccentricity-dependent analyses showed a U-shaped decoding curve, confirming that these results are not explained by spillover of peripheral activity or large receptive fields. Finally, fluctuations in foveal decodability covaried with activity in the intraparietal sulcus, thus providing a candidate region for driving foveal feedback. These findings suggest that foveal cortex predicts the features of incoming stimuli through feedback from higher cortical areas, which offers a candidate mechanism underlying stable perception.

## Introduction

Human vision relies heavily on foveal processing. Even though this region of the retina covers only a fraction of the visual field, it occupies a disproportionately large portion of neurons in the early visual cortex (*Curcio et al., 1990*; *Curcio and Allen, 1990*; *Hendrickson, 2005*; *Schira et al., 2009*). To take advantage of the fovea's high resolution, humans perform several rapid eye movements per second, with each saccade bringing an object of interest into the fovea (*O'Regan, 1992*; *Carpenter, 2000*). Despite these frequent disruptions and dramatic shifts in retinal input, our perception of the visual environment remains stable, coherent, and continuous (*Golomb and Mazer, 2021*; *Melcher and Colby, 2008*; *Burr et al., 1994*). This perceptual stability, while often taken for granted, points toward sophisticated neural mechanisms that integrate visual information across separate gaze fixations (*Wurtz, 2008*; *Cavanagh et al., 2010*; *Merriam et al., 2007*; *Denagamage et al., 2024*).

In the domain of visual perception, predictions may critically contribute to maintaining perceptual stability across rapid shifts in gaze (*Rao and Ballard, 1999*; *de Lange et al., 2018*; *Clark, 2013*). One compelling hypothesis suggests that perceptual continuity arises through predictive feedback of peripheral information from higher cortical areas to foveal retinotopic regions. This feedback informs foveal regions about the expected visual features of a stimulus prior to its direct fixation to prepare for the shift in visual input (*Kroell and Rolfs, 2022*). Support for this foveal-prediction hypothesis comes from psychophysical studies, which find that features of peripheral saccade targets are enhanced in the presaccadic fovea (*Kroell and Rolfs, 2022*; *Kroell and Rolfs, 2025*).

Current evidence for the role of feedback in foveal prediction is indirect, relying mostly on the interpretation of behavioral reports. While multiple studies have shown feedback of peripheral information to the fovea during fixation (*Williams et al., 2008*; *Fan et al., 2016*; *Costantino et al., 2025*; *Stewart et al., 2020*), this effect was only shown when features of the target stimulus were task relevant. The relationship of these foveal feedback signals to foveal prediction during saccade preparation has remained unclear. Furthermore, the nature of such feedback signals and whether they resemble activity patterns elicited by direct foveal stimulation has remained poorly understood.

In this study, we directly tested saccade-related foveal feedback in the brain to address (1) whether feedback is specific to stimulus features and which features are fed back, (2) whether feedback activation resembles activation elicited by direct stimulus presentation, and (3) which brain regions mediate this effect. To systematically address these open questions, we developed a gaze-contingent functional magnetic resonance imaging (fMRI) paradigm that allowed us to disentangle neural activity attributable to direct visual input from activity exclusively related to foveal feedback. By removing peripheral saccade targets before participants could fixate them, we ensured that observed neural activation within foveal retinotopic regions must originate from feedback rather than direct foveal stimulation. Furthermore, by employing naturalistic stimuli whose visual shape and semantic content were independently manipulated, we could explicitly assess the specificity and content of the signals fed back to early foveal cortex.

Our findings robustly demonstrate the presence of feedback in early visual areas, including primary visual cortex, indicated by reliable decoding of peripheral saccade targets from foveal retinotopic areas, despite the absence of direct foveal stimulation. Critically, this decoding was selective for stimulus shape and not influenced by semantic category, indicating a predominantly low-to-mid-level visual representation. Eccentricity-dependent analyses showed a U-shaped decoding curve, demonstrating that these results cannot be explained by spillover of peripheral activity or large receptive fields. Cross-decoding analyses further confirmed the similarity between feedback and direct visual representations in the fovea, reinforcing the shared nature of neural codes. Finally, exploratory analyses identified the intraparietal sulcus (IPS), a brain area integral to visuomotor coordination and eye movement planning, as a likely candidate involved in driving foveal feedback. These findings reveal a plausible neural implementation for perceptual continuity and may also facilitate object recognition across saccades (*Herwig and Schneider, 2014*; *Blom et al., 2020*).

## Results

Given the fast time scale of saccade-related processes, it is challenging to investigate foveal feedback using fMRI, which is based mostly on a sluggish hemodynamic response. To address this challenge and dissociate neural processes elicited by direct visual input from those related to foveal feedback, we

designed a gaze-contingent functional MRI study where the saccade target was removed before it could reach the central 2 degrees of visual angle (dva) of the fovea (1 dva radius), from which we decoded. To maximise statistical power, we implemented the paradigm using a block design (*Figure 1A*). During a block, participants fixated a fixation point until a stimulus appeared in the periphery, cueing them to execute a saccade toward the stimulus. As soon as their gaze came within 6.5 dva of the saccade target, the stimulus disappeared, ensuring that no part of the stimulus ever appeared in the central 2 dva of the fovea. On subsequent trials of the block, the same stimulus appeared in the participant's periphery to cue another saccade, a process which was repeated until the end of a block. The presentation time of each stimulus depended on the saccade latency (M=240.70 ms). While the fovea is commonly defined as the central 5 dva of the visual field (*Curcio and Allen, 1990*; *Hendrickson, 2009*; *Hendrickson, 2005*), we focused here on the central part of the fovea, commonly referred to as the foveola (*Poletti et al., 2017*; *Heckenlively and Arden, 2006*), within 2 dva (1 dva radius), to avoid any overlap of the stimulus and the area from which we decoded. This narrow definition also allowed for enough space to execute a saccade without the stimulus breaching this region. *Figure 1C* shows that this paradigm was successful in preventing the target from reaching the central 2 dva of the fovea in 99.27% of saccades. All blocks in which the target could have appeared in this region were removed from further analysis (see methods section quantification and statistical analysis).

To compare the experimental condition to activation elicited by direct foveal input, we included a control condition in which participants were instructed to fixate a point at the center of the screen, with the stimulus appearing directly in the center of the fovea (*Figure 1A*, bottom). To keep visual stimulation frequency comparable, the timing of stimulus appearance and disappearance was recorded for each participant in the experimental condition and replayed in the control condition. To elucidate the content of the stimulus information fed back to foveal retinotopic areas, we used four different natural stimuli (one stimulus per block) allowing us to disentangle the nature of the representation in the foveal retinotopic cortex (*Figure 1B*). The stimuli were manipulated to match in either shape (horizontal vs. vertical) or semantic category (animals vs. instruments).

## Decoding foveal feedback

To test for the presence of stimulus-specific activation in foveal regions of early visual cortex, we used cross-validated multivariate decoding (*Hebart and Baker, 2018*), which reveals information that allows discriminating between different stimuli. We compared all pairs of stimuli in the experimental and control conditions, respectively (chance level: 50%). In the experimental condition, we found above-chance decoding in central foveal V1 (t(27) = 8.81, *p*<0.001, mean = 57.43%) (*Figure 2A*). This result demonstrates that information about the peripheral saccade targets is present in central foveal regions of V1, despite never appearing in the corresponding part of the fovea. To compare this finding to direct foveal stimulus presentation, we repeated the same analysis for the control condition, where we also found strong significant decoding (t(27) = 19.92, *p*<0.001, mean = 84.06%) (*Figure 2A*). To further examine the nature of the neural representation elicited by foveal feedback, we cross-decoded from experimental to control trials by training a classifier on data from central foveal V1 in the experimental condition and testing it on the control condition. Decoding was significantly above chance (t(27) = 5.22, *p*<0.001, mean = 57.2%), indicating a similar representational format between the neural representation elicited by direct presentation of the stimulus in the fovea and that elicited by foveal feedback (*Figure 2A*).

This pattern of results may alternatively be explained by spillover from peripheral regions or large receptive fields in the fovea reaching into the periphery (cf *Williams et al., 2008*). To address this issue, we investigated decoding as a function of eccentricity in early visual regions (*Figure 2B*). Foveal decoding due to peripheral spillover would predict a monotonic relationship between peripheral and foveal decoding in the experimental condition. Instead, we found a U-shaped relationship, with stronger decoding from peripheral and foveal regions compared to parafoveal regions. We tested this relationship using a weighted quadratic regression and found significant positive curvature for decoding in all early visual areas (V1: t(27) = 3.98, *p*=0.008, V2: t(27) = 3.03, *p*=0.02, V3: t(27) = 2.776, *p*=0.025, one-sided). These results highlight the spatial pattern of foveal feedback, separating decoding due to direct stimulus presentation in the periphery and decoding due to feedback to foveal regions. As expected, in the control condition, decoding was the highest in the center of gaze and dropped off towards the periphery.

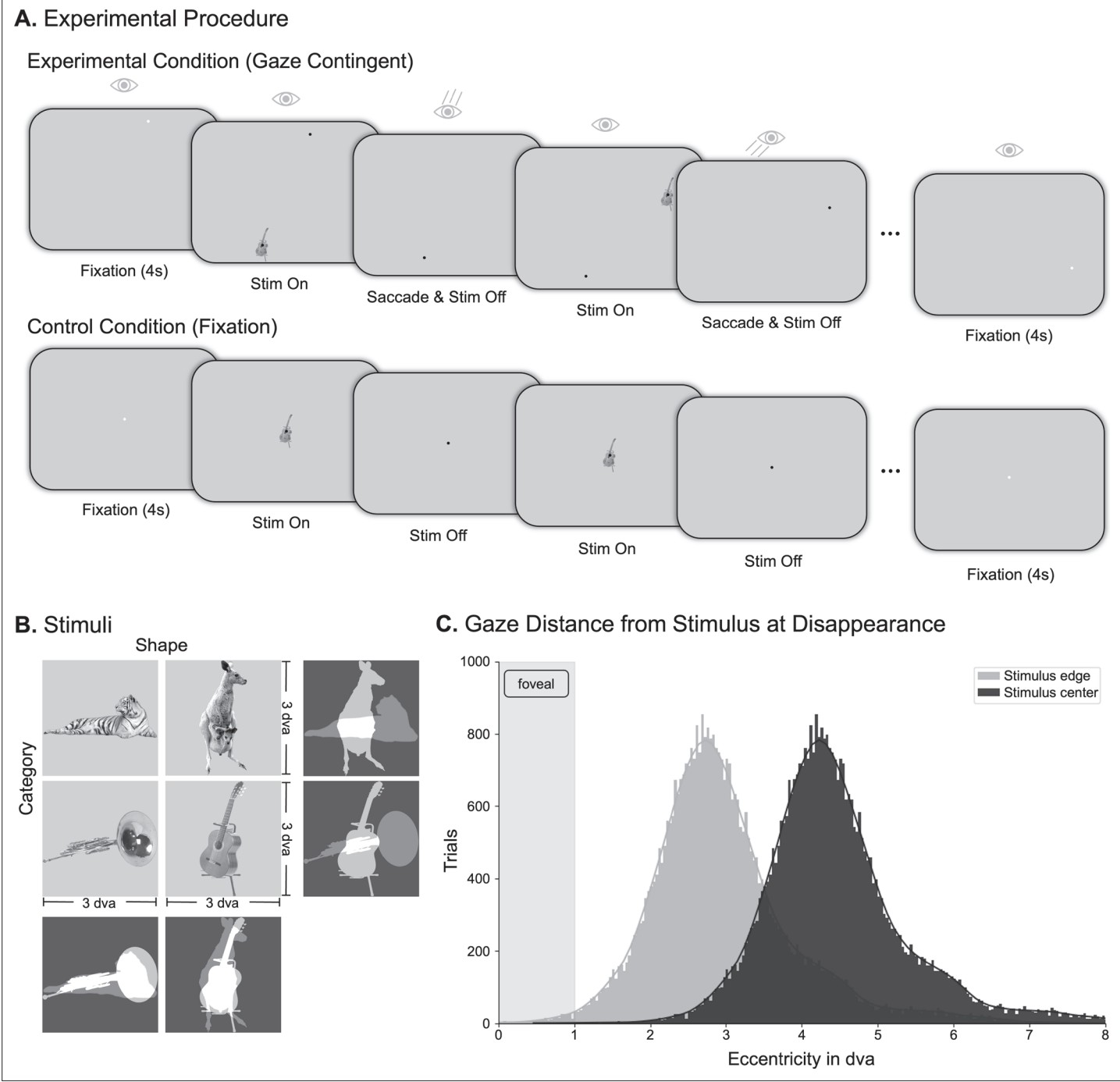

**Figure 1.** Experimental setup. (**A**) Illustration of one block of the experimental and control conditions, respectively. During each block in the experimental condition, a peripheral saccade target was shown, which participants were instructed to fixate. The target disappeared before it could be foveated, and once fixation was achieved, a new target appeared, until the block was over (duration: 11 s). The timing of target appearance and disappearance from the experimental condition was recorded and used in the control condition, where targets appeared at fixation. (**B**) Depiction of the four stimuli used in the experiment, which were matched in either visual shape (horizontal/vertical) or semantic category (animal/instrument). Each stimulus appeared equally often in both conditions. (**C**) Histogram of the gaze distance from the stimulus right after stimulus disappearance, including all trial from all 28 participants . The blocks in which the stimulus edge may have appeared in the participants' fovea during at least one saccade were excluded.

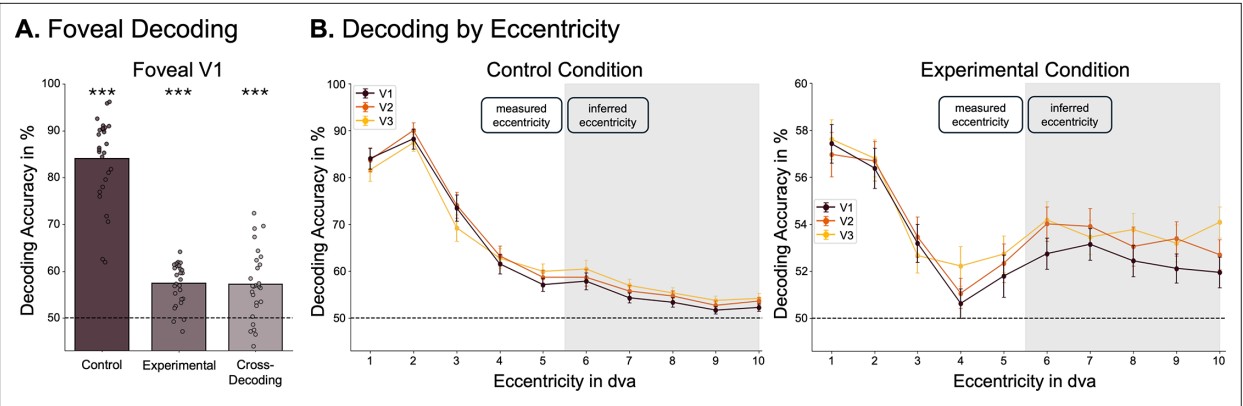

**Figure 2.** Foveal feedback can be decoded from V1. (**A**) Average decoding accuracy over all pairwise comparisons for control (t(27) = 19.92, *p*<0.001, mean = 84.06%) and experimental (t(27) = 8.81, *p*<0.001, mean = 57.43%) conditions and for cross-decoding from experimental to control condition (t(27) = 5.22, *p*<0.001, mean = 57.2%). (**B**) Average decoding accuracies for all early visual areas as a function of eccentricity for both experimental and control conditions. Error bars represent standard error of the mean. Note that the graphs have different scales of decoding accuracy. The central eccentricities (1–5 dva) were measured using a retinotopic localizer, the outer ones (6–10 dva) were inferred from structural data using Neuropythy (*Benson and Winawer, 2018*).

## Foveal feedback is sensitive to stimulus shape, not semantic category

Our use of natural stimuli allowed us to test the effects of shape and category on decoding accuracy to better understand the nature of the information fed back to foveal areas (*Figure 3A*). Decoding was assessed between visually similar yet semantically dissimilar stimuli (across category), between semantically similar yet visually dissimilar stimuli (across shape), and between visually and semantically dissimilar stimuli (across both). The latter comparison served as a baseline, assessing how good decoding is for maximally different stimuli.

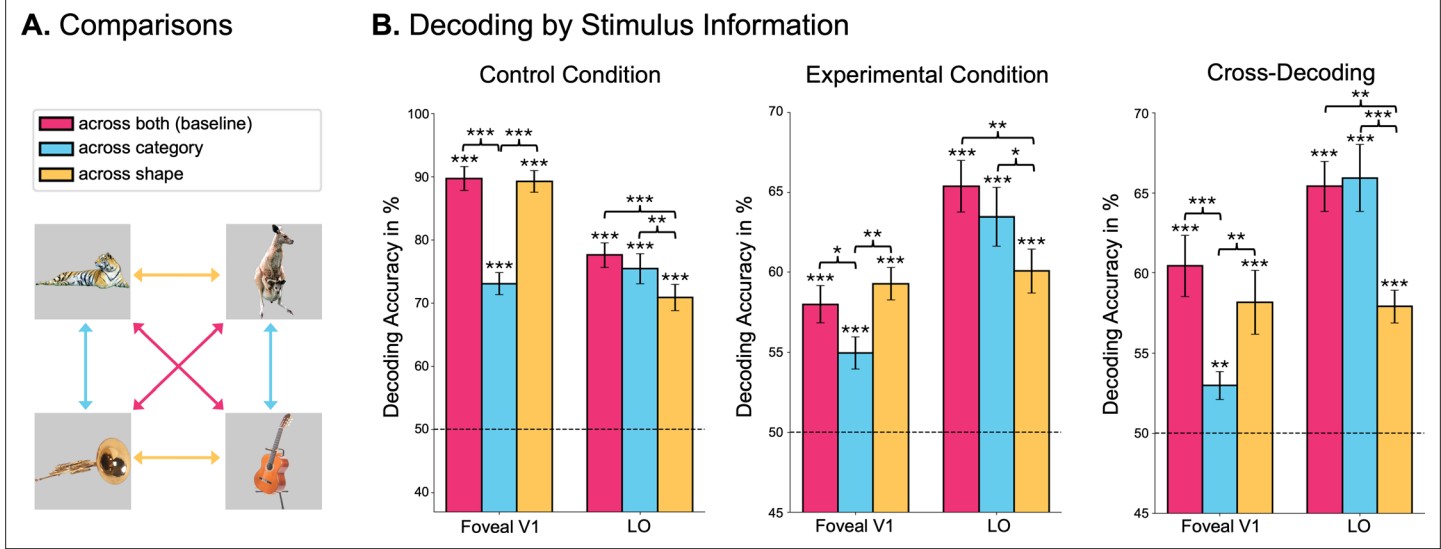

**Figure 3.** Foveal feedback is sensitive to stimulus shape, not semantic category. (**A**) Schematic depiction of comparisons to assess the information content of the neural representations. Comparisons across categories assess similarity of representations in terms of visual stimulus properties, with lower decoding accuracies indicating coding for visual information. Similarly, comparing across shape assesses categorical information. Comparing across both serves as a baseline, showing how high decoding accuracy is between maximally different stimuli. (**B**) Decoding accuracies for all comparisons using data from foveal regions of V1 and from the lateral occipital area (LO) (n=28). Error bars represent the standard error from the mean. Note that the graphs have different y-axes of decoding accuracy.

The online version of this article includes the following figure supplement(s) for figure 3:

**Figure supplement 1.** Decoding stimulus content from all early foveal areas.

A similar pattern of decoding accuracies emerged in both experimental and control conditions: Decoding from foveal V1 across category dropped significantly relative to baseline (experimental condition: t(27) = 2.25, p=0.033, difference = 3.03%; control condition: t(27) = 14.74, p<0.001, difference = 16.64%), while decoding across shape remained high (**Figure 3B**). This pattern indicates that the nature of the feedback signal in the experimental condition was related to stimulus shape information and not semantic category. The fact that the overall pattern of results across conditions looks similar in the experimental and control conditions is in line with the notion that direct stimulus presentation and foveal feedback elicit similar neural representation, as suggested by the cross-decoding results described above (**Figure 2A**). To test the degree to which the classifier was able to pick up on category information, we repeated the same analysis in the lateral occipital area (LO), which has been shown to capture higher-level information about object category (**Grill-Spector et al., 2001**). In this area, the pattern was reversed: Decoding dropped across shape relative to baseline (experimental condition: t(27) = 3.41, p=0.002, difference = 5.31%; control condition: t(27) = 7.25, p<0.001, difference = 6.7%), while it remained high across category, which suggests that this activation more strongly reflects information about the semantic category than stimulus shape.

## The role of IPS in mediating foveal feedback

While the previous analyses revealed the pattern of foveal feedback in early visual regions, they left open which neural regions might be involved in driving or mediating this effect. To this end, we conducted an exploratory analysis looking at the block-by-block fluctuations in foveal decodability in the experimental condition. As a measure of decodability, we used the continuous decision value of the classifier, which signifies the distance to the classifier's hyperplane on a given trial. Using a parametric modulation analysis, we explored which brain region's activity increased or decreased as a function of foveal decodability. To control for the effects of the direct peripheral presentation of the stimulus, we used the block-by-block fluctuations of peripheral decoding in the experimental condition as a baseline.

Since foveal feedback is a process tightly linked to saccadic eye movements (**Kroell and Rolfs, 2022**), we hypothesized that regions associated with eye movements are most likely involved in driving this effect. We focused on three regions that have consistently been associated with eye movements and object representations: Frontal eye fields (FEF) (**Paus, 1996**; **Vernet et al., 2014**), intraparietal sulcus (IPS) (**Andersen et al., 1992**; **Andersen, 1989**), and lateral occipital area (LO)

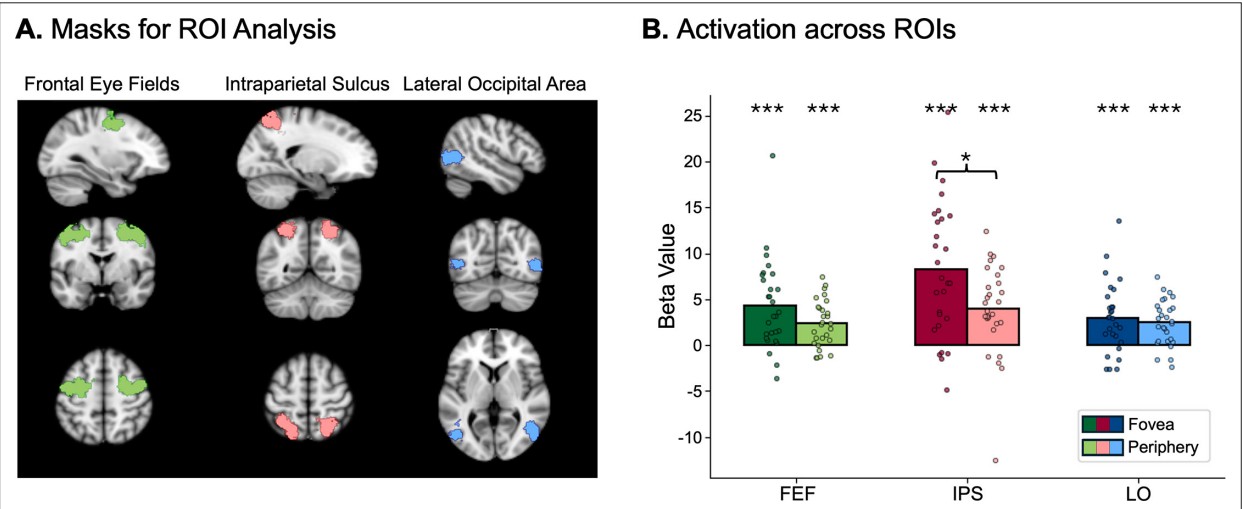

**Figure 4.** Correlation of foveal decoding with region of interest (ROI) activation. (**A**) Cortical masks used in the ROI analyses. These masks were generated using Neurosynth (**Yarkoni et al., 2011**) with the keyword 'eye movements.' For later analyses, only the 100 most activating voxels were selected in each area. (**B**) Results of the ROI analyses comparing neural activation as a function of foveal and peripheral decoding in three key areas related to eye movements (n=28). Activation in the intraparietal sulcus (IPS) was significantly higher in association with foveal decoding compared to peripheral decoding (t(27) = 2.53, p=0.026, difference = 4.22).

The online version of this article includes the following figure supplement(s) for figure 4:

**Figure supplement 1.** Parametric modulation analysis in the control condition.

(**Kawawaki et al., 2006**; **Figure 4A**). Not all voxels in these regions were expected to be functionally active. Therefore, to focus analyses on the most relevant voxels, we selected the 100 voxels in each of these regions that activated most strongly in the experimental condition in general. The region of interest (ROI) analyses showed that all of these areas were significantly related to both foveal and peripheral decoding in the experimental condition (**Figure 4B**). Specifically, IPS showed significantly larger activation related to foveal decoding compared to peripheral decoding (t(27) = 2.53, p=0.026, difference = 4.22), suggesting that IPS may be involved in foveal feedback. While effects in the other regions went in the expected direction, they remained non-significant after correcting for multiple comparisons (LO: t(27) = 0.67, p=0.767, difference = 0.53; FEF: t(27) = 2.07, p=0.072, difference = 1.98). We conducted the same analysis in the control condition, which showed a significant decrease in FEF (t(27) = –4.62, p<0.001, difference = 10.36) and IPS (t(27) = –3.61, p=0.004, difference = 11.6) and an increase in LO (t(27) = 5.11, p<0.001, difference = 16.89) in association with foveal decoding compared to peripheral decoding. In contrast, we found no significant association between peripheral decoding and any of the ROIs (FEF: t(27) = 0.49, p=1.0; IPS: t(27) = 1.35, p=0.56; LO: t(27) = 2.43, p=0.07). These findings confirm that our main effects were not simply explained by global brain fluctuations or signal-to-noise ratio, since under those conditions, we would have expected a similar relationship between foveal and peripheral decoding as in the experimental condition (**Figure 4— figure supplement 1**).

## Discussion

The present preregistered study provides evidence that early foveal retinotopic areas are involved in the processing of peripheral saccade targets, even if the stimuli are never presented in the central part of the fovea. Using a combination of fMRI and eye-tracking, we were able to decode the identity of saccade targets from foveal regions of early visual cortex (**Figure 2A**), suggesting that shape-specific information about peripheral targets is fed back to foveal areas during saccade preparation. We also showed that this foveal feedback is unlikely to be caused by spillover from peripheral regions (**Figure 2B**) and that it has a similar neural representation as direct stimulus presentation, as shown by the above-chance cross-decoding (**Figure 2A**). This cross-decoding alleviates concerns about the type of information picked up by the decoder in the experimental condition. That is, it cannot be explained by the decoder picking up small changes in eye movements between different stimuli (**Williams et al., 2008**), since the representation - at least in part - transfers to the control condition, where participants did not move their eyes. Furthermore, we showed that foveal feedback is sensitive to the shape, but not the semantic category of the stimulus (**Figure 3B**), which suggests that foveal feedback in early visual cortex is rather rudimentary and does not convey a category-invariant stimulus representation. These findings are in line with recent work showing that feedback to primary visual cortex in a fixation condition may primarily carry low-level perceptual information (**Costantino et al., 2025**). Lastly, in an exploratory analysis, we identified the intraparietal sulcus (IPS) as a candidate region for driving foveal feedback (**Figure 4B**).

### Foveal feedback during saccade preparation

While previous studies have shown that peripheral information can be decoded from foveal regions of the visual cortex and can affect foveal processing, this effect has primarily been studied during passive fixation and under specific task conditions related to distinctive spatial features of the peripheral stimulus (**Williams et al., 2008**; **Costantino et al., 2025**; **Fan et al., 2016**; **Yu and Shim, 2016**; **Chambers et al., 2013**). In the absence of target-specific tasks, or using a different control task, foveal feedback was not observed (**Williams et al., 2008**; **Knapen et al., 2016**; **Fan et al., 2016**). The present study employed a paradigm in which participants performed a saccade towards the target—irrespective of target features or a specific task on the target. Despite the absence of a target-specific task, we still observed robust decoding from foveal regions. This task independence is reminiscent of the saccade-based recruitment of feature-based attention in area V4 of the macaque brain (**Burrows et al., 2014**), showing that neurons tuned to the features of an imminent saccade target increase their responsiveness. However, in contrast to such spatially invariant feature selection, our results show clear spatial selectivity for foveal locations. Furthermore, we found a specific positive association of IPS activity with the experimental condition, not the control condition, which is in line with the idea that the

foveal feedback effect reported in this study is related to saccade preparation (*Curtis and Connolly, 2008*; *Gaymard et al., 1998*). Since humans invariably perform saccades to bring relevant objects into foveal view, instead of scrutinizing them peripherally, foveal feedback during fixation (e.g. *Williams et al., 2008*) could possibly be the result of the preparation of eye movements that are not executed (cf. *Kroell and Rolfs, 2022*; *Fan et al., 2016*; *Chambers et al., 2013*; *Yu and Shim, 2016*). The function of predictive foveal feedback in this context would be to support continuity of visual processing across eye movements that routinely change objects' locations in retinotopic coordinates (cf. *Kroell and Rolfs, 2022*). Indeed, covert attention and saccade preparation are strongly coupled processes (*Li et al., 2021*; *Kowler et al., 1995*; *Deubel and Schneider, 1996*; *Montagnini and Castet, 2007*; *Rolfs and Carrasco, 2012*; *Rolfs et al., 2011*). The task used by *Williams et al., 2008* would, in natural vision, likely involve an eye movement to the peripheral targets. While these findings offer a plausible alternative understanding of the results of *Williams et al., 2008*, this interpretation remains speculative, and more research is needed to determine whether our findings and theirs result from the same underlying mechanism.

## Saccadic remapping or foveal prediction

Saccadic remapping, that is, the increase of activity of neurons in anticipation of a stimulus entering their receptive field, has been observed all over the visual cortex (*Golomb and Mazer, 2021*; *Duhamel et al., 1992*; *Merriam et al., 2007*; *Mirpour and Bisley, 2012*), including primary visual cortex (*Nakamura and Colby, 2002*; *Knapen et al., 2016*). While this effect has been widely reported, there is little evidence that saccadic remapping also encodes feature information in humans (*Rao and Ballard, 1999*; *Xiao et al., 2024*; *Knapen et al., 2009*; *Knapen et al., 2010*; *Lescroart et al., 2016*; *Yao et al., 2016*), but see *Denagamage et al., 2024*. One exception is the presaccadic integration of features across two peripheral locations, provided they are the current and future location of an attended stimulus (*Harrison et al., 2013*; *Szinte et al., 2015*). Such integration, however, could be explained by presaccadic updating of spatial attention pointers (*Rolfs et al., 2011*) that link two retinotopic locations to one object, rather than remapping of feature information per se (*Pelli and Cavanagh, 2013*), although this is a topic of ongoing discussion (*Golomb and Mazer, 2021*). This explanation does not apply to the effect observed in the present study, since there never was a stimulus presented in the fovea, so remapping of spatial attention to the fovea (as in *Rolfs et al., 2011*, Figure 5) would not suffice to explain the data.

Foveal prediction, on the other hand, genuinely involves the transfer of information of saccade target features. In their psychophysical experiments, *Kroell and Rolfs, 2022*; *Kroell and Rolfs, 2025* found that, during saccade preparation, features of the peripheral saccade target were enhanced in the pre-saccadic fovea. In contrast, three independent studies found no automatic selection of saccade target features at peripheral locations (*Born et al., 2013*; *Jonikaitis and Theeuwes, 2013*; *White et al., 2013*). Thus, while saccade-based feature-based attention is evident in visual cortex (*Burrows et al., 2014*), it is not sufficient to explain feature predictions before saccades. These findings led (*Kroell and Rolfs, 2022*; *Kroell and Rolfs, 2025*) to conclude that the fovea plays a unique role in maintaining perceptual continuity by predicting future inputs during saccade preparation. Our results support this view by showing shape-sensitive decoding, as well as cross-decoding from experimental to control condition, indicating that foveal feedback may lead early foveal regions to share features with the peripheral target stimulus in anticipation of an upcoming saccade. In line with this, *Lescroart et al., 2016* found no evidence for periphery-to-periphery feature remapping, and *Chiu and Golomb, 2025* found supporting evidence for remapping of object-location binding from periphery to fovea but not periphery to periphery. Together, these findings suggest that foveal processing is uniquely equipped for predicting feature information of upcoming stimuli.

## Comparing foveal feedback to direct presentation

The decoding patterns in the present study revealed that the information about the saccade targets that is fed back to the foveal cortex may reflect shape information but does not contain higher-level categorical information. These decoding patterns resemble the ones we found in the control condition, where the stimuli were presented directly in the fovea (*Figure 3B*). Lastly, it was possible to cross-decode by training a decoder on foveal V1 data from the experimental condition and decoding stimulus identity from the same regions in the control condition at above chance level (*Figure 2A*).

These results are in line with behavioral studies showing that presenting a foveal foil stimulus identical to the peripheral target shortly after target onset improves visual discrimination of the peripheral target (*Yu and Shim, 2016*). Together, these findings suggest shared representational formats in early visual areas between foveal prediction and direct stimulus presentation, which indicates that foveal feedback reflects low-to-mid-level features of the target, similarly to the direct presentation of the stimulus.

## IPS as a candidate modulator of foveal prediction

In a parametric modulation analysis, we found that the intraparietal sulcus (IPS) was significantly more active in association with foveal decoding compared to peripheral decoding in the experimental condition (*Figure 4B*). This area has been described as neither a strictly visual nor motor area but instead as performing visuomotor integration functions, such as determining the spatial location of saccade targets and forming plans to make eye movements (*Andersen et al., 1992*; *Andersen, 1989*). Further research has shown that this region represents salient stimuli, relative to the center of gaze (*Colby and Duhamel, 1996*). This integrative function makes the IPS an ideal candidate for modulating feature-specific feedback to foveal areas during saccade preparation. While this analysis is exploratory, it offers yet another indication that foveal feedback is inherently linked to saccadic eye movements, and that IPS could play an important role in driving this effect. Future hypothesis-driven research could specifically target this region to more clearly determine its role in foveal feedback.

## Limitations and future directions

Despite the insights gained in the present study, several open questions remain. We did not specifically test whether we can find foveal feedback from peripheral targets without any stimulus-specific task (e.g. no eye movements), which is relevant to showing that foveal feedback is task-dependent. However, similar control conditions have been run by *Williams et al., 2008* and *Knapen et al., 2016*. Neither found any stimulus-specific foveal activation with peripheral target presentation in the absence of a target-specific task. Additionally, while the stimulus-specific effects reported in the present study were robust, the results were limited to four different stimuli, since the addition of further conditions would have led to a reduction in statistical power. Future studies could expand upon the present approach by increasing the number of stimuli, possibly collecting data across multiple sessions to achieve sufficiently large effects. Furthermore, while using an fMRI block-design paradigm allowed us to conduct precise spatial analyses of different retinotopic regions with high statistical power, it did not allow for any temporal analyses. Under these circumstances, we cannot fully rule out that the observed effects were influenced by working memory (*Harrison and Tong, 2009*) or mental imagery (*Albers et al., 2013*). However, given previous psychophysical work (*Kroell and Rolfs, 2022*) and the fact that stimulus features were not task-relevant, participants had no incentive to engage in these processes, making it unlikely that they played a strong role in our findings. Another limitation of our work is that, while our results are consistent with psychophysical studies defining the temporal onset of foveal feedback during saccade preparation (*Kroell and Rolfs, 2025*; *Fan et al., 2016*), we cannot rule out that post-saccadic processes might have also influenced the observed effects (*Chambers et al., 2013*). Lastly, an additional condition in which participants make a saccade to a neighboring stimulus could elucidate if foveal prediction is exclusive to the target of a saccade. While these questions offer exciting research avenues for future studies, our results demonstrate the importance of foveal feedback during saccadic eye movements, offering a plausible candidate mechanism for our ability to integrate visual information across saccades. They also pave the way for future research about how foveal prediction may facilitate object recognition by giving object processing a head start before fixation onset.

## Materials and methods

This study was pre-registered on OSF (https://osf.io/rxacd/), detailing the hypotheses, methodology, and planned analyses prior to data collection, with the exception of the exploratory parametric modulation analysis. While the preregistration was initiated and written before conducting the study, due to an error on the authors' side, it was only submitted after submission of the manuscript. However,

timestamps on the website document demonstrate that the preregistration text was complete before data collection and not altered afterwards.

The dataset collected for this study is available on OpenNeuro: https://doi.org/10.18112/open-neuro.ds005933.v1.0.0. The experimental and analysis code are available on GitHub: https://github.com/Lucakaemmer/FovealFeedback (copy archived at *Kaemmer, 2026*).

## Experimental model and study participant details

The experiment was performed with human participants recruited by the recruitment system of the Max Planck Institute for Cognitive and Brain Sciences in Leipzig, where the experiment was conducted. Thirty participants were tested, which is similar to comparable studies. Two participants were excluded. One person had poor eye-tracking performance, which impaired the functioning of the gaze-contingent aspect of the experiment (fewer than 100 valid saccades in the experiment). The other was excluded based on poor fMRI data, likely due to drowsiness, which made it impossible to generate retinotopic masks. The remaining 28 participants (14 male and 14 female) were 18–40 years old (average 27.54 years), healthy, right-handed, and had normal or corrected-to-normal vision. Participants were compensated with €12 per hour. The experiment was conducted in accordance with the declaration of Helsinki, and the experimental procedure was approved by the ethics review board of the University of Leipzig (reference number: 421/23-ek). The participants gave written informed consent before taking part in the study.

## Method details

### Stimuli

The stimuli were four different object images taken from the Hemera object dataset (H. Technologies. Hemera photo objects). These stimuli were chosen to match either regarding their visual shape (horn and tiger, guitar and kangaroo) or their semantic category (horn and guitar, tiger, and kangaroo). The stimuli that were visually similar were also semantically dissimilar, and vice versa (*Figure 1B*). All stimuli were normalized regarding their overall luminance and root mean square contrast and were presented at a size of 3 dva.

### Behavioral task

In the experimental condition, objects appeared in eight different locations arranged in a circle around the screen, each of them 4 dva away from the center of the screen (*Figure 1A*). Experimental blocks started with a white fixation point appearing in one of these eight locations. After 4 s, the fixation point turned black, and one of the four stimuli appeared in one of the three locations on the opposite side of the screen, 7.4 dva or 8 dva away from the fixation point. The stimulus only appeared if participants were fixating on the fixation point. Stimulus appearance was the cue for the participant to perform a saccade. As soon as their gaze came within 6.5 dva of the stimulus during the saccade, the stimulus disappeared to prevent it from appearing in the participant's fovea. After an inter-trial interval of 0.5 s, the same stimulus appeared again at the next location. This pattern repeated for 11 s, concluding one experimental block. One run consisted of 20 blocks, switching to a different stimulus in each block. The experiment included five experimental runs and five control runs.

In the control condition, stimuli were presented at the center of the fovea, with fixation at the center of the screen (*Figure 1A*). The overall timing of the stimulus presentation was the same as in the experimental condition. To this end, the timing of stimulus appearance and disappearance in the experimental condition, which depended on the participants' eye movements, was recorded and used in the control condition to provide the same exposure duration to the stimulus across conditions.

Lastly, an eccentricity retinotopic localizer and standard object localizer task were used to create retinotopic mapping and to identify object-processing areas in each participant. The retinotopic localizer consisted of six iterations of contracting or expanding rings to map each participant's eccentricity in early visual areas. The object localizer consisted of blocks of either object images or scrambled images to separate object-processing areas from lower-level visual areas.

### Imaging

fMRI scanning was performed at the Max Planck Institute for Human Cognitive and Brain Sciences in Leipzig, Germany, using a Siemens Magnetom Skyra 3T scanner equipped with a 32-channel head

coil (Siemens, Erlangen). Whole-brain T2*-weighted echo-planar images (EPI) were collected with the following parameters: repetition time (TR)=2000 ms, echo time (TE)=23.6 ms, flip angle = 80°, field of view (FOV)=204 mm², voxel size = 2×2×2 mm³, and 69 transverse slices with no gap. An interleaved slice acquisition was used with a multiband acceleration factor of 3 and no in-plane acceleration, and the phase encoding direction was anterior to posterior. High-resolution T1-weighted anatomical images were acquired at the end of the scanning session using a standard magnetization-prepared rapid gradient echo (MPRAGE) sequence. Parallel imaging with GRAPPA was utilized for the T1-weighted MPRAGE sequence with an acceleration factor of 2.

For eye tracking, we used an EyeLink 1000+ Eyetracker, sampling the right eye at 1000 Hz. The eye tracker was calibrated using a 13-point setup to ensure high eye tracking quality. The stimuli were presented using a VPixx PROPixx projector, set to a real-time refresh rate of 180 Hz to allow for the gaze-contingent disappearance of the stimulus before participants could fixate on it. The presentation was programmed in Python using PsychoPy (*Peirce et al., 2019*).

Each participant completed one scanning session, starting with 10 experimental runs (five gaze-contingent and five fixation runs), which took 320 s each. The retinotopic localizer run took 206 s, and the object localizer run took 426 s, bringing the overall time inside the MRI scanner to around 80 min, including structural scans, breaks, and the calibration of the eye tracker.

## Quantification and statistical analysis

The fMRI data were preprocessed using fMRIPrep version 20.2.0 (*Esteban et al., 2019*). Brain surfaces were reconstructed using recon-all from FreeSurfer v6.0.1 (*Dale et al., 1999*). Functional data were slice-time corrected using 3dTshift from AFNI v16.2.07 (*Cox, 1996*) and motion-corrected using mcflirt (*Jenkinson et al., 2012*). The functional data used for the decoding analysis remained in each participant's native space, without registering to a brain template.

Univariate analyses were conducted on all experimental and control runs using FSL's FEAT (v6.00) to perform a general linear model analysis (*Jenkinson et al., 2012*). The resulting beta values for each block were then used in the subsequent multivariate decoding analysis. We chose pairwise classification over multiclass classification (1) since our intent was not to build a classifier for real-world applications but to infer from the presence of above-chance accuracies the presence of discriminative information (*Hebart and Baker, 2018*) and (2) since most common multiclass classification approaches implicitly rely on the comparison of multiple pairwise classifiers anyway. For the decoding analyses and statistical tests, a custom-written Python pipeline was used; a linear support vector machine with fivefold cross-validation was used for decoding, provided by the Python package sklearn (*Pedregosa et al., 2011*). Decoding accuracies were compared with t-tests using the Python package SciPy (*Virtanen et al., 2020*). One-sample t-tests were used to assess if decoding accuracies were significantly different from chance. Since a within-subjects design was used in which all participants went through all the conditions, paired, two-sided t-tests were used to compare decoding accuracies (across shape, across category, across both). The significance threshold was set at $p<0.05$. Significance levels in figures are indicated as follows: *$p<0.05$, **$p<0.01$, ***$p<0.001$.

Anatomical masks for V1, V2, and V3 were generated for each participant using Neuropythy (*Benson and Winawer, 2018*). The same package was used to estimate the outer retinotopic eccentricities from 6 to 10 dva. The inner eccentricities from 1 to 5 degrees were estimated using the functional retinotopic localizer. A univariate analysis was used to estimate the voxels responding to a certain range of eccentricity. Similarly, anatomical masks for LO were generated for each participant using the functional object localizer.

The parametric modulation analysis was conducted with FSL's FEAT. Using the decoders' block-by-block fluctuations in decision scores as regressors, the parametric modulation analysis was run on the preprocessed data from the experimental runs, which was converted to MNI space to allow for second-level analyses across participants. Masks for the eye-tracking ROIs were generated using the website Neurosynth (*Yarkoni et al., 2011*), which synthesizes data from thousands of fMRI studies to create functional cortical masks in MNI space. Using the keyword 'eye movements,' we generated masks of regions known to be associated with eye movements: Frontal eye fields (FEF) (*Paus, 1996*; *Vernet et al., 2014*), intraparietal sulcus (IPS) (*Andersen et al., 1992*; *Andersen, 1989*), and lateral occipital area (LO) (*Kawawaki et al., 2006*; *Figure 4A*). We then isolated the 100 voxels per area that activated most strongly in the experimental condition overall, independent of which stimulus was

shown. Beta estimates for these 100 voxels in each area were then compared with paired, one-sided t-tests using SciPy to test our hypothesis that foveal decoding was associated with larger neural activation in our ROIs. These tests were corrected for multiple comparisons using Bonferroni correction.

To assure that there was no direct foveal stimulation in the experimental condition, we excluded all the experimental blocks in which any part of the stimulus may have appeared in the participants' fovea during at least one saccade, which happened in 0.73% of all saccades. Since entire blocks were removed when they contained a single such saccade, this led to an exclusion of 8.49% of experimental blocks. To keep the balance between experimental and control conditions, we also excluded the corresponding blocks from the control condition.

The processing of offline eye movement data was performed in Matlab 2018b (Mathworks, Natick, MA, USA). Within an experimental block, a trial only started (characterized by the appearance of a stimulus) once participants fixated within 1 dva of the initial fixation point and only ended once they fixated within 1 dva of the target fixation point. In the offline analysis, a saccade was detected when 2D velocity exceeded 5 standard deviations from the median for a minimum of 5 ms (*Engbert and Mergenthaler, 2006*). Saccade candidates that were less than 10 ms apart usually resulting from post-saccadic oscillations (*Schweitzer and Rolfs, 2022*) were merged into a single saccade. On average, participants performed 1084 (±206.17) trials over the course of the experiment, which amounts to 10.84 trials per experimental block. Overall, participants reached the target area (within 2 dva of stimulus center) in a single saccade in 76.07% of all stimulus appearances, and in 16.94% in more than one saccade. Note that humans can plan two saccades in advance and still allocate attention to the final goal ahead of the first saccade (*Rolfs et al., 2011*; *Baldauf and Deubel, 2008*; *Gersch et al., 2004*; *Godijn and Theeuwes, 2003*). In 6.99% of trials, no saccade was detected in the offline analysis, probably due to tracking noise in the MRI scanner. In the worst case, these trials would only have added noise. Average saccade latency was 240.70 ms, and average saccade amplitude was 6.76 dva. All the figures were prepared using Python and PowerPoint.

## Acknowledgements

We would like to thank the members of the Vision and Computational Cognition Group at the Max Planck Institute for Human Cognitive and Brain Sciences for their valuable feedback and contributions to this study. We would also like to thank the staff in the MRI department at the MPI-CBS that supported the data collection. This work was supported by a doctoral student scholarship awarded to LK by the German Academic Scholarship Foundation ('Studienstiftung des Deutschen Volkes'), a research group grant by the Max Planck Society awarded to MNH, the ERC Starting Grant project COREDIM (ERC-StG-2021–101039712) and the Hessian Ministry of Higher Education, Science, Research and Art (LOEWE Start Professorship to MNH and Excellence Program 'The Adaptive Mind'). MR was supported by the ERC Consolidator Grant project VIS-A-VIS (grant agreement 865715). The funders had no role in study design, data collection and analysis, decision to publish or preparation of the manuscript. Open access funding provided by Max Planck Society.

## Additional information

### Funding

| Funder | Grant reference number | Author |
|---|---|---|
| German National Academic Foundation | | Luca Kämmer |
| Max Planck Society | | Martin N Hebart |
| European Research Council | 10.3030/101039712 | Martin N Hebart |
| European Research Council | 10.3030/865715 | Martin Rolfs |

| Funder | Grant reference number | Author |
|--------|------------------------|--------|

The funders had no role in study design, data collection and interpretation, or the decision to submit the work for publication. Open access funding provided by Max Planck Society.

## Author contributions

Luca Kämmer, Conceptualization, Data curation, Formal analysis, Investigation, Visualization, Methodology, Writing – original draft; Lisa M Kroell, Conceptualization, Formal analysis; Tomas Knapen, Methodology, Writing – review and editing; Martin Rolfs, Conceptualization, Supervision, Writing – review and editing; Martin N Hebart, Conceptualization, Resources, Supervision, Funding acquisition, Methodology, Project administration, Writing – review and editing

## Author ORCIDs

Luca Kämmer ⓘ https://orcid.org/0009-0009-8046-5724
Lisa M Kroell ⓘ https://orcid.org/0000-0002-3508-5214
Martin Rolfs ⓘ https://orcid.org/0000-0002-8214-8556
Martin N Hebart ⓘ https://orcid.org/0000-0001-7257-428X

## Ethics

Human subjects: The experiment was conducted in accordance with the declaration of Helsinki, and the experimental procedure was approved by the ethics review board of the University of Leipzig (reference number: 421/23-ek). The participants gave written informed consent before taking part in the study.

Reviewer #2 (Public review): https://doi.org/10.7554/eLife.107053.3.sa1
Reviewer #3 (Public review): https://doi.org/10.7554/eLife.107053.3.sa2
Author response https://doi.org/10.7554/eLife.107053.3.sa3

---

# Additional files

## Supplementary files

MDAR checklist

## Data availability

The dataset collected for this study is available on OpenNeuro: https://doi.org/10.18112/openneuro.ds005933.v1.0.0. The experimental and analysis code are available on GitHub: https://github.com/Lucakaemmer/FovealFeedback (copy archived at *Kaemmer, 2026*).

The following dataset was generated:

| Author(s) | Year | Dataset title | Dataset URL | Database and Identifier |
|-----------|------|---------------|-------------|-------------------------|
| Kaemmer L, Hebart M | 2025 | Foveal Feedback | https://doi.org/10.18112/openneuro.ds005933.v1.0.0 | OpenNeuro, 10.18112/openneuro.ds005933.v1.0.0 |

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
