## [Editor Report · eLife Assessment]

This **valuable** study addresses a question related to how we achieve visual stability across saccadic eye movements. The authors' gaze-contingent fMRI design provides **convincing** evidence that peripherally presented visual stimuli are represented in foveal visual cortex prior to a saccade. The results will be of interest to vision scientists and behavioural neuroscientists.

---

## [Referee Report · Reviewer #2 (Public review)]

Summary:

This study investigated whether the identity of a peripheral saccade target object is fed back to the foveal retinotopic cortex during saccade preparation, a critical prediction of the foveal prediction hypothesis proposed by Kroell & Rolfs (2022). To achieve this, the authors leveraged a gaze-contingent fMRI paradigm, where the peripheral saccade target was removed before the eyes landed near it, and used multivariate decoding analysis to quantify identity information in the foveal cortex. The results showed that the identity of the saccade target object can be decoded based on foveal cortex activity, despite the fovea never directly viewing the object, and that the foveal feedback representation was similar to passive viewing and not explained by spillover effects. Additionally, exploratory analysis suggested IPS as a candidate region mediating such foveal decodability. Overall, these findings provide neural evidence for the foveal cortex processing the features of the saccade target object, potentially supporting the maintenance of perceptual stability across saccadic eye movements.

Strengths:

This study is well-motivated by previous theoretical findings (Kroell & Rolfs, 2022), aiming to provide neural evidence for a potential neural mechanism of trans-saccadic perceptual stability. The question is important, and the gaze-contingent fMRI paradigm is a solid methodological choice for the research goal. The use of stimuli allowing orthogonal decoding of stimulus category vs stimulus shape is a nice strength, and the resulting distinctions in decoded information by brain region are clean. The results will be of interest to readers in the field, and they fill in some untested questions regarding pre-saccadic remapping and foveal feedback.

Weaknesses:

The authors have done a nice job addressing the previous weaknesses. The remaining weaknesses / limitations are appropriately discussed in the manuscript. E.g., the use of only 4 unique stimuli in the experiment. The findings are intriguing and relevant to saccadic remapping and foveal feedback, but somewhat limited in terms of the ability to draw theoretical distinctions between these related phenomena.

Specifics:

The revised manuscript is much improved in terms of framing and discussion of the prior literature, and the theoretical claims are now stated with appropriate nuance.

I have two remaining minor suggestions/comments, which the authors may optionally respond to:

(1) In the parametric modulation analysis, the authors' additional analyses nicely addresses my concern and strengthens the claim. However, the description in the revised manuscript (Pg 7 Ln 190-191) is minimal and may be difficult to grasp what the control analysis is about and how it rules out alternative explanations to the IPS findings. The authors may wish to elaborate on the description in the text.

(2) Out of curiosity (not badgering): The authors argued that the findings of Harrison et al. (2013) and Szinte et al. (2015) can be explained by feature integration between the currently attended location and its future, post-saccadic location. Couldn't the same argument apply in the current paradigm, where attention at the saccade target gets remapped to the pre-saccadic fovea (see also Rolfs et al., 2011 Fig 5), thus leading to the observed feature remapping?

---

## [Referee Report · Reviewer #3 (Public review)]

Summary:

In this paper the authors used fMRI to determine whether peripherally-viewed objects could be decoded from foveal cortex, even when the objects themselves were never viewed foveally. Specifically they investigated whether pre-saccadic target attributes (shape, semantic category) could be decoded from foveal cortex. They found that object shape, but not semantic category could be decoded, providing evidence that foveal feedback relies on low-mid-level information. The authors claim that this provides evidence for a mechanism underlying visual stability and object recognition across saccades.

Strengths:

I think this is another nice demonstration that peripheral information can be decoded from / is processed in foveal cortex - the methods seem appropriate, and the experiments and analyses carefully conducted, and the main results seem convincing. The paper itself was very clear and well-written.

Weaknesses:

Given that foveal feedback has been found in previous studies that don't incorporate saccades, it is still unclear how this mechanism might specifically contribute to stability across saccades, rather than just being a general mechanism that aids the processing/discrimination of peripherally-viewed stimuli. The authors address this point, but I guess whether foveal feedback during fixation and saccade prep are really the same, is ultimately a question that needs more experimental work to disentangle.

---

## [Author Response]

The following is the authors’ response to the original reviews.

**Public Reviews:**

**Reviewer #1 (Public review):**
Summary:The main contributions of this paper are: (1) a replication of the surprising prior finding that information about peripherally-presented stimuli can be decoded from foveal V1 (Williams et al 2008), (2) a new demonstration of cross-decoding between stimuli presented in the periphery and stimuli presented at the fovea, (3) a demonstration that the information present in the fovea is based on shape not semantic category, and (4) a demonstration that the strength of foveal information about peripheral targets is correlated with the univariate response in the same block in IPS.Strengths:The design and methods appear sound, and finding (2) above is new, and importantly constrains our understanding of this surprising phenomenon. The basic effect investigated here is so surprising that even though it has been replicated several times since it was first reported in 2008, it is useful to replicate it again.

We thank the reviewer for their summary. While we agree with many points, we would like to respectfully push back on the notion that this work is a replication of Williams et al. (2008). What our findings share with those of Williams is a report of surprising decoding at the fovea without foveal stimulation. Beyond this similarity, we treat these as related but clearly separate findings, for the following reasons:

(1) Foveal feedback, as shown by Williams et al. (2008) and others during fixation, was only observed during a shape discrimination task, specific to the presented stimulus. Control experiments without such a task (or a color-related task) did not show effects of foveal feedback. In contrast, in the present study, the participants’ task was merely to perform saccades towards stimuli, independently of target features. We thus show that foveal feedback can occur independently of a task related to stimulus features. This dissociation demonstrates that our study must be tapping into something different than reported by Williams.

(2) In a related study, Kroell and Rolfs (2022, 2025) demonstrated a connection between foveal feedback and saccade preparation, including the temporal details of the onset of this effect before saccade execution, highlighting the close link of this effect to saccade preparation. Here we used a very similar behavioral task to capture this saccade-related effect in neural recordings and investigate how early it occurs and what its nature is. Thus, there is a clear motivation for this study in the context of eye movement preparation that is separate from the previous work by Williams.

(3) Lastly, decoding in the experimental task was positively associated with activity in FEF and IPS, areas that have been reliably linked to saccade preparation. We have now also performed an additional analysis (see our response to Specific point 2 of Reviewer 2) showing that decoding in the control condition did not show the same association, further supporting the link of foveal feedback to saccade preparation.

Despite our emphasis on these critical differences in studies, covert peripheral attention, as required by the task in Williams et al., and saccade preparation in natural vision, as in our study, are tightly coupled processes. Indeed, the task in Williams et al. would, during natural vision, likely involve an eye movement to the peripheral target. While speculative, a parsimonious and ecologically valid explanation is that both ours and earlier studies involve eye movement preparation, for which execution is suppressed, however, in studies enforcing fixation (e.g., Williams et al., 2008). We now discuss this idea of a shared underlying mechanism more extensively in the revised manuscript (pg 8 ln 228-240).

Weaknesses:(1) The paper, including in the title ("Feedback of peripheral saccade targets to early foveal cortex") seems to assume that the feedback to foveal cortex occurs in conjunction with saccade preparation. However, participants in the original Williams et al (2008) paper never made saccades to the peripheral stimuli. So, saccade preparation is not necessary for this effect to occur. Some acknowledgement and discussion of this prior evidence against the interpretation of the effect as due to saccade preparation would be useful. (e.g., one might argue that saccade preparation is automatic when attending to peripheral stimuli.)

We agree that the effects Williams et al. showed were not sufficiently discussed in the first version of this manuscript. To more clearly engage with these findings we now introduce saccade related foveal feedback (foveal prediction) and foveal feedback during fixation separately in the introduction (pg 2 ln 46-59).

We further added another section in the discussion called “Foveal feedback during saccade preparation” in which we discuss how our findings are related to Williams et al. and how they differ (pg 8 ln 211-240).

As described in our previous response, we believe that our findings go beyond those described by Williams et al. (2008) and others in significant ways. However, during natural vision, the paradigm used by Williams et al. (2008) would likely be solved using an eye movement. Thus, while participants in Williams et al. (2008) did not execute saccades, it appears plausible that they have prepared saccades. Given the fact that covert peripheral attention and saccade preparation are tightly coupled processes (Kowler et al., 1995, Vis Res; Deubel & Schneider, 1996, Vis Res; Montagnini & Castet, 2007, J Vis; Rolfs & Carrasco, 2012, J Neurosci; Rolfs et al., 2011, Nat Neurosci), their results are parsimoniously explained by saccade preparation (but not execution) to a behaviorally relevant target.

(2) The most important new finding from this paper is the cross-decodability between stimuli presented in the fovea and stimuli presented in the periphery. This finding should be related to the prior behavioral finding (Yu & Shim, 2016) that when a foveal foil stimulus identical to a peripheral target is presented 150 ms after the onset of the peripheral target, visual discrimination of the peripheral target is improved, and this congruency effect occurred even though participants did not consciously perceive the foveal stimulus (Yu, Q., & Shim, W. M., 2016). Modulating foveal representation can influence visual discrimination in the periphery (Journal of Vision, 16(3), 15-15).

We thank the reviewer for highlighting this highly relevant reference. In the revised version of the manuscript, we now put more emphasis on the finding of cross-decodability (pg 2 ln 60-61). We now also discuss Yu et al.’s finding, which support our conclusion that foveal feedback and direct stimulus presentation share representational formats in early visual areas (pg 9 ln 277-279).

(3) The prior literature should be laid out more clearly. For example, most readers will not realize that the basic effect of decodability of peripherally-presented stimuli in the fovea was first reported in 2008, and that that original paper already showed that the effect cannot arise from spillover effects from peripheral retinotopic cortex because it was not present in a retinotopic location between the cortical locus corresponding to the peripheral target and the fovea. (For example, this claim on lines 56-57 is not correct: "it remains unknown (1) whether information is fed back all the way to early visual areas"). What is needed is a clear presentation of the prior findings in one place in the introduction to the paper, followed by an articulation and motivation of the new questions addressed in this paper. If I were writing the paper, I would focus on the cross-decodability between foveal and peripheral stimuli, as I think that is the most revealing finding.

We agree that the structure of the introduction did not sufficiently place our work in the context of prior literature. We have now expanded upon our Introduction section to discuss past studies of saccade- and fixation-related foveal feedback (pg 2 ln 49-59), laying out how this effect has been studied previously. We also removed the claim that "it remains unknown (1) whether information is fed back all the way to early visual areas", where our intention was to specifically focus on foveal prediction. We realize that this was not clear and hence removed this section. Instead, we now place a stronger focus on the cross-decodability finding (pg 2 ln 60-61).

**Reviewer #2 (Public review):**
Summary:This study investigated whether the identity of a peripheral saccade target object is predictively fed back to the foveal retinotopic cortex during saccade preparation, a critical prediction of the foveal prediction hypothesis proposed by Kroell & Rolfs (2022). To achieve this, the authors leveraged a gaze-contingent fMRI paradigm, where the peripheral saccade target was removed before the eyes landed near it, and used multivariate decoding analysis to quantify identity information in the foveal cortex. The results showed that the identity of the saccade target object can be decoded based on foveal cortex activity, despite the fovea never directly viewing the object, and that the foveal feedback representation was similar to passive viewing and not explained by spillover effects. Additionally, exploratory analysis suggested IPS as a candidate region mediating such foveal decodability. Overall, these findings provide neural evidence for the foveal cortex processing the features of the saccade target object, potentially supporting the maintenance of perceptual stability across saccadic eye movements.Strengths:This study is well-motivated by previous theoretical findings (Kroell & Rolfs, 2022), aiming to provide neural evidence for a potential neural mechanism of trans-saccadic perceptual stability. The question is important, and the gaze-contingent fMRI paradigm is a solid methodological choice for the research goal. The use of stimuli allowing orthogonal decoding of stimulus category vs stimulus shape is a nice strength, and the resulting distinctions in decoded information by brain region are clean. The results will be of interest to readers in the field, and they fill in some untested questions regarding pre-saccadic remapping and foveal feedback.

We thank the reviewer for the positive assessment of our study.

Weaknesses:The conclusions feel a bit over-reaching; some strong theoretical claims are not fully supported, and the framing of prior literature is currently too narrow. A critical weakness lies in the inability to test a distinction between these findings (claiming to demonstrate that "feedback during saccade preparation must underlie this effect") and foveal feedback previously found during passive fixation (Williams et al., 2008). Discussions (and perhaps control analysis/experiments) about how these findings are specific to the saccade target and the temporal constraints on these effects are lacking. The relationship between the concepts of foveal prediction, foveal feedback, and predictive remapping needs more thorough treatment. The choice to use only 4 stimuli is justified in the manuscript, but remains an important limitation. The IPS results are intriguing but could be strengthened by additional control analysis. Finally, the manuscript claims the study was pre-registered ("detailing the hypotheses, methodology, and planned analyses prior to data collection"), but on the OSF link provided, there is just a brief summary paragraph, and the website says "there have been no completed registrations of this project".

We thank the reviewer for these helpful considerations. We agree that some of the claims were not sufficiently supported by the evidence, and in the revised manuscript, we added nuance to those claims (pg 8 ln 211-240). Furthermore, we now address more directly the distinction between foveal feedback during fixation and foveal feedback (foveal prediction) during saccade preparation. In particular, we now describe the literature about these two effects separately in the introduction (pg 2 ln 46-59), and we have added a new section in the discussion (“Foveal feedback during saccade preparation”) that more thoroughly explains why a passive fixation condition would have been unlikely to produce the same results we find (pg 8 ln 211-227). We also adapted the section about “Saccadic remapping or foveal prediction”, clearly delineating foveal prediction from feature remapping and predictive updating of attention pointers. As recommended by the reviewer, we conducted the parametric modulation analyses on the control condition, strengthening the claim that our findings are saccade-related. These results were added as Supplementary Figure 2 and are discussed in (pg 7 ln 190-191) and (pg 8 ln 224-227).

Lastly, we would like to apologize about a mistake we made with the pre-registration. We realized that the pre-registration had indeed not been submitted. We have now done so without changing the pre-registration itself, which can be seen from the recent activity of the preregistration (screenshot attached in the end). After consulting an open science expert at the University of Leipzig, we added a note of this mistake to the methods section of the revised manuscript (pg 10 ln 326-332). We could remove reference to this preregistration altogether, but would keep it at the discretion of the editor.

Specifics:(1) In the eccentricity-dependent decoding results (Figure 2B), are there any statistical tests to support the results being a U-shaped curve? The dip isn't especially pronounced. Is 4 degrees lower than the further ones? Are there alternative methods of quantifying this (e.g., fitting it to a linear and quadratic function)?

We statistically tested the U-shaped relationship using a weighted quadratic regression, which showed significant positive curvature for decoding between fovea and periphery in all early visual areas (V1: t(27) = 3.98, p = 0.008, V2: t(27) = 3.03, p = 0.02, V3: t(27) = 2.776, p = 0.025, one-sided). We now report these results in the revised manuscript (pg 5 ln 137-138).

(2) In the parametric modulation analysis, the evidence for IPS being the only region showing stronger fovea vs peripheral beta values was weak, especially given the exploratory nature of this analysis. The raw beta value can reflect other things, such as global brain fluctuations or signal-to-noise ratio. I would also want to see the results of the same analysis performed on the control condition decoding results.

We appreciate the reviewer’s suggestion and repeated the same parametric modulation analysis on the control condition to assess the influence of potential confounds on the overall beta values (Supplementary Figure 2). The results show a negative association between foveal decoding and FEF and IPS (likely because eye movements in the control condition lead to less foveal presentation of the stimulus) and a positive association with LO. Peripheral decoding was not associated with significant changes in any of the ROIs, indicating that global brain fluctuations alone are not responsible for the effects reported in the experimental condition. The results of this analysis thus show a specific positive association of IPS activity with the experimental condition, not the control condition, which is in line with the idea that the foveal feedback effect reported in this study may be related to saccade preparation.

(3) Many of the claims feel overstated. There is an emphasis throughout the manuscript (including claims in the abstract) that these findings demonstrate foveal prediction, specifically that "image-specific feedback during saccade preparation must underlie this effect." To my understanding, one of the key aspects of the foveal prediction phenomenon that ties it closely to trans-saccadic stability is its specificity to the saccade target but not to other objects in the environment. However, it is not clear to what degree the observed findings are specific to saccade preparation and the peripheral saccade target. Should the observers be asked to make a saccade to another fixation location, or simply maintain passive fixation, will foveal retinotopic cortex similarly contain the object's identity information? Without these control conditions, the results are consistent with foveal prediction, but do not definitively demonstrate that as the cause, so claims need to be toned down.

We fully agree with the reviewer and toned down claims about foveal prediction. We engage with the questions raised by the reviewer more thoroughly in the new discussion section “Foveal feedback during saccade preparation”.

In addition, we agree that another condition in which subjects make a saccade towards a different location would have been a great addition that we also considered, but due to concerns with statistical power did not add. While including such a condition exceeds the scope of the current study, we included this limitation in the Discussion section (pg 10 ln 316) and hope that future studies will address this question.

(4) Another critical aspect is the temporal locus of the feedback signal. In the paradigm, the authors ensured that the saccade target object was never foveated via the gaze-contingent procedure and a conservative data exclusion criterion, thus enabling the test of feedback signals to foveal retinotopic cortex. However, due to the temporal sluggishness of fMRI BOLD signals, it is unclear when the feedback signal arrives at the foveal retinotopic cortex. In other words, it is possible that the feedback signal arrives after the eyes land at the saccade target location. This possibility is also bolstered by Chambers et al. (2013)'s TMS study, where they found that TMS to the foveal cortex at 350-400 ms SOA interrupts the peripheral discrimination task. The authors should qualify their claims of the results occurring "during saccade preparation" (e.g., pg 1 ln 22) throughout the manuscript, and discuss the importance of temporal dynamics of the effect in supporting stability across saccades.

We fully agree that the sluggishness of the fMRI signal presents an important challenge in investigating foveal feedback. We have now included this limitation in the discussion (pg 10 ln 306-318). We also clarify that our argument connects to previous studies investigating the temporal dynamics of foveal feedback using similar tasks (pg 10 ln 313-316). Specifically, in their psychophysical work, Kroell and Rolfs (2022) and (2025) showed that foveal feedback occurs before saccade execution with a peak around 80 ms before the eye movement.

(5) Relatedly, the claims that result in this paradigm reflect "activity exclusively related to predictive feedback" and "must originate from predictive rather than direct visual processes" (e.g., lines 60-65 and throughout) need to be toned down. The experimental design nicely rules out direct visual foveal stimulation, but predictive feedback is not the only alternative to that. The activation could also reflect mental imagery, visual working memory, attention, etc. Importantly, the experiment uses a block design, where the same exact image is presented multiple times over the block, and the activation is taken for the block as a whole. Thus, while at no point was the image presented at the fovea, there could still be more going on than temporally-specific and saccade-specific predictive feedback.

We agree that those claims could have misled the reader. Our intention was to state that the activation originates from feedback rather than direct foveal stimulation because of the nature of the design. We have now clarified these statements (pg 2 ln 65) and also included a discussion of other effects including imagery and working memory in the limitations section (pg 10 ln 306-313).

(6) The authors should avoid using the terms foveal feedback and foveal prediction interchangeably. To me, foveal feedback refers to the findings of Williams et al. (2008), where participants maintained passive fixation and discriminated objects in the periphery (see also Fan et al., 2016), whereas foveal prediction refers to the neural mechanism hypothesized by Kroell & Rolfs (2022), occurring before a saccade to the target object and contains task irrelevant feature information.

We agree, and we have now adopted a clearer distinction between these terms, referring to foveal prediction only when discussing the distinct predictive nature of the effect discovered by Kroell and Rolfs (2022). Otherwise we referred to this effect as foveal feedback.

(7) More broadly, the treatment of how foveal prediction relates to saccadic remapping is overly simplistic. The authors seem to be taking the perspective that remapping is an attentional phenomenon marked by remapping of only attentional/spatial pointers, but this is not the classic or widely accepted definition of remapping. Within the field of saccadic remapping, it is an ongoing debate whether (/how/where/when) information about stimulus content is remapped alongside spatial location (and also whether the attentional pointer concept is even neurophysiologically viable). This relationship between saccadic remapping and foveal prediction needs clarification and deeper treatment, in both the introduction and discussion.

We thank the reviewer for their remarks. We reformulated the discussion section on “Saccadic remapping or foveal prediction” to include the nuances about spatial and feature remapping laid out in the reviewer’s comment (pg 8-9 ln 241-269). We also put a stronger focus on the special role the fovea seems to be playing regarding the feedback of visual features (pg 8-9 ln 265-269).

(8) As part of this enhanced discussion, the findings should be better integrated with prior studies. E.g., there is some evidence for predictive remapping inducing integration of non-spatial features (some by the authors themselves; Harrison et al., 2013; Szinte et al., 2015). How do these findings relate to the observed results? Can the results simply be a special case of non-spatial feature integration between the currently attended and remapped location (fovea)? How are the results different from neurophysiological evidence for facilitation of the saccade target object's feature across the visual field (Burrow et al., 2014)? How might the results be reconciled with a prior fMRI study that failed to find decoding of stimulus content in remapped responses (Lescroart et al, 2016)? Might this reflect a difference between peripheral-to-peripheral vs peripheral-to-foveal remapping? A recent study by Chiu & Golomb (2025) provided supporting evidence for peripheral-to-fovea remapping (but not peripheral-to-peripheral remapping) of object-location binding (though in the post-saccadic time window), and suggested foveal prediction as the underlying mechanism.

We thank the reviewer for raising these intriguing questions. We now address them in the revised discussion. We argue that the findings by Harrison et al., 2013 and Szinte et al., 2015 of presaccadic integration of features across two peripheral locations can be explained by presaccadic updating of spatial attention pointers rather than remapping of feature information (pg 8 ln 248-253). The lack of evidence for periphery-to-periphery remapping (Lescroart et al, 2016) and the recent study by Chiu & Golomb (2025) showing object-location binding from periphery to fovea nicely align with our characterization of foveal processing as unique in predicting feature information of upcoming stimuli (pg 8-9 ln 265-269). Finally, we argue that the global (i.e., space-invariant) selection task-irrelevant saccadic target features (Burrows et al., 2014) is well-established at the neural level, but does not suffice to explain the spatially specific nature of foveal prediction (pg 8 ln 220-224). We now include these studies in the revised discussion section.

**Reviewer #3 (Public review):**
Summary:In this paper, the authors used fMRI to determine whether peripherally viewed objects could be decoded from the foveal cortex, even when the objects themselves were never viewed foveally. Specifically, they investigated whether pre-saccadic target attributes (shape, semantic category) could be decoded from the foveal cortex. They found that object shape, but not semantic category, could be decoded, providing evidence that foveal feedback relies on low-mid-level information. The authors claim that this provides evidence for a mechanism underlying visual stability and object recognition across saccades.Strengths:I think this is another nice demonstration that peripheral information can be decoded from / is processed in the foveal cortex - the methods seem appropriate, and the experiments and analyses are carefully conducted, and the main results seem convincing. The paper itself was very clear and well-written.

We thank the reviewer for this positive evaluation of our work. As discussed in our response to Reviewer 1, we now elaborate on the differences between previous work showing decoding of peripheral information from foveal cortex from the effect shown here. While there are important similarities between these findings, foveal prediction in our study occurs in a saccade condition and in the absence of a task that is specific to stimulus features.

Weaknesses:There are a couple of reasons why I think the main theoretical conclusions drawn from the study might not be supported, and why a more thorough investigation might be needed to draw these conclusions.(1) The authors used a blocked design, with each object being shown repeatedly in the same block. This meant that the stimulus was entirely predictable on each block, which weakens the authors' claims about this being a predictive mechanism that facilitates object recognition - if the stimulus is 100% predictable, there is no aspect of recognition or discrimination actually being tested. I think to strengthen these claims, an experiment would need to have unpredictable stimuli, and potentially combine behavioural reports with decoding to see whether this mechanism can be linked to facilitating object recognition across saccades.

We appreciate the reviewer’s point and would like to highlight that it was not our intention to claim a behavioral effect on object recognition. We believe that an ambiguous formulation in the original abstract may have been interpreted this way, and we thus removed this reference. We also speculated in our Discussion that a potential reason for foveal prediction could be a headstart in peripheral object recognition and in the revised manuscript more clearly highlight that this is a potential future direction only.

(2) Given that foveal feedback has been found in previous studies that don't incorporate saccades, how is this a mechanism that might specifically contribute to stability across saccades, rather than just being a general mechanism that aids the processing/discrimination of peripherally-viewed stimuli? I don't think this paper addresses this point, which would seem to be crucial to differentiate the results from those of previous studies.

We fully agree that this point had not been sufficiently addressed in the previous version of the manuscript. As described in our responses to similar comments from reviewers 1 and 2, we included an additional section in the Discussion (“Foveal feedback during saccade preparation”) to more clearly delineate the present study from previous findings of foveal feedback. Previous studies (Williams et al., 2008) only found foveal feedback during narrow discrimination tasks related to spatial features of the target stimulus, not during color-discrimination or fixation-only tasks, concluding that the observed effect must be related to the discrimination behavior. In contrast, we found foveal feedback (as evidenced by decoding of target features) during a saccade condition that was independent of the target features, suggesting a different role of foveal feedback than hypothesized by Williams et al. (2008).

**Recommendations for the authors:**

**Reviewer #2 (Recommendations for the authors):**
(A) Minor comments:(1) The task should be clarified earlier in the manuscript.

We now characterise the task in the abstract and clarified its description in the third paragraph, right after introducing the main literature.

(2) Is there actually only 0.5 seconds between saccades? This feels very short/rushed.

The inter-trial-interval was 0.5 seconds, though effectively it varied because the target only appeared once participants fixated on the fixation dot. Note that this pacing is slower than the rate of saccades in natural vision (about 3 to 4 saccades per second).Participants did not report this paradigm as rushed.

(3) Typo on pg2 ln64 (whooe).

Fixed.

(4) Can the authors also show individual data points for Figures 3 and 4?

We added individual data points for Figures 4 and S2

(5) The MNI coordinates on Figure 4A seem to be incorrect.

We took out those coordinates.

(6) Pg4 ln126 and pg6 ln194, why cite Williams et al. (2008)?

We included this reference here to acknowledge that Williams et al. raised the same issues. We added a “cf.” before this reference to clarify this.

(7) Pg7 ln207 Fabius et al. (2020) showed slow post-saccadic feature remapping, rather than predictive remapping of spatial attention.

We have corrected this mistake.

(8) The OSF link is valid, but I couldn't find a pre-registration.

The issue with the OSF link has been resolved. The pre-registration had been set up but not published. We now published it without changing the original pre-registration (see the screenshot attached).

(9) I couldn't access the OpenNeuro repository.

The issue with the OpenNeuro link has been resolved.

(B) Additional references you may wish to include:(1) Burrows, B. E., Zirnsak, M., Akhlaghpour, H., Wang, M., & Moore, T. (2014). Global selection of saccadic target features by neurons in area v4. Journal of Neuroscience.(2) Chambers, C. D., Allen, C. P., Maizey, L., & Williams, M. A. (2013). Is delayed foveal feedback critical for extra-foveal perception?. Cortex.(3) Chiu, T. Y., & Golomb, J. D. (2025). The influence of saccade target status on the reference frame of object-location binding. Journal of Experimental Psychology. General.(4) Harrison, W. J., Retell, J. D., Remington, R. W., & Mattingley, J. B. (2013). Visual crowding at a distance during predictive remapping. Current Biology.(5) Lescroart, M. D., Kanwisher, N., & Golomb, J. D. (2016). No evidence for automatic remapping of stimulus features or location found with fMRI. Frontiers in Systems Neuroscience.(6) Moran, C., Johnson, P. A., Hogendoorn, H., & Landau, A. N. (2025). The representation of stimulus features during stable fixation and active vision. Journal of Neuroscience.(7) Szinte, M., Jonikaitis, D., Rolfs, M., Cavanagh, P., & Deubel, H. (2016). Presaccadic motion integration between current and future retinotopic locations of attended objects. Journal of Neurophysiology.

We thank the reviewer for pointing out these references. We have included them in the revised version of the manuscript.

**Reviewer #3 (Recommendations for the authors):**
I just have a few minor points where I think some clarifications could be made.(1) Line 64 - "whooe" should be "whoose" I think.

Fixed.

(2) Around line 53 - you might consider citing this review on foveal feedback - https://doi.org/10.1167/jov.20.12.2

We included the reference (pg 2 ln 55).

(3) Line 129 - you mention a u-shaped relationship for decoding - I wasn't quite sure of the significance/relevance of this relationship - it would be helpful to expand on this / clarify what this means.

We have expanded this section and added statistical tests of the u-shaped relationship in decoding using a weighted quadratic regression. We found significant positive curvature in all early visual areas between fovea and periphery (V1: t(27) = 3.98, p = 0.008, V2: t(27) = 3.03, p = 0.02, V3: t(27) = 2.776, p = 0.025). These findings support a u-shaped relationship. We now report these results in the revised manuscript (pg 5 ln 137-138).

(4) Figure 1 - it would be helpful to indicate how long the target was viewed in the "stim on" panels - I assume it was for the saccade latency, but it would be good to include those values in the main text.

We included that detail in the text (pg 3 ln 96-97).